# Labour-Market Characteristics and Self-Rated Health: Evidence from the China Health and Retirement Longitudinal Study

**DOI:** 10.3390/ijerph20064748

**Published:** 2023-03-08

**Authors:** Yuwei Pan, Hynek Pikhart, Martin Bobak, Jitka Pikhartova

**Affiliations:** Research Department of Epidemiology and Public Health, University College London, 1–19 Torrington Place, London WC1E 7HB, UK

**Keywords:** aging, self-rated health, labour market, disease prevention, China

## Abstract

In the face of labour-force ageing, understanding labour-market characteristics and the health status of middle-aged and older workers is important for sustainable social and economic development. Self-rated health (SRH) is a widely-used instrument to detect health problems and predict mortality. This study investigated labour-market characteristics that may have an impact on the SRH among Chinese middle-aged and older workers, using data from the national baseline wave of the China Health and Retirement Longitudinal Study. The analytical sample included 3864 individuals who at the time held at least one non-agricultural job. Fourteen labour-market characteristics were clearly defined and investigated. Multiple logistic regression models of the associations of each labour-market characteristic with SRH were estimated. Seven labour-market characteristics were associated with higher odds of poor SRH when controlled for age and sex. Employment status and earned income remained significantly associated with poor SRH, when controlling for all the sociodemographic factors and health behaviours. Doing unpaid work in family businesses is associated with 2.07 (95% CI, 1.51–2.84) times probability of poor SRH, compared with employed individuals. Compared with more affluent individuals (highest quintile of earned income), people in the fourth and fifth quintiles had 1.92 (95% CI, 1.29–2.86) times and 2.72 (95% CI, 1.83–4.02) times higher chance, respectively, of poor SRH. In addition, residence type and region were important confounders. Measures improving adverse working conditions should be taken to prevent future risk of impaired health among the Chinese middle-aged and older workforce.

## 1. Introduction

As a result of rapid population ageing, middle-aged and older workers are becoming the main part of the labour force in China [1,2]. According to the Seventh National Population Census in China in 2020, workers aged 45 years or more accounted for 43.2% of the total working population [3]. Thus understanding the characteristics of middle-aged and older workers, and maintaining their health status, are both important for the sustainable development of China’s society. According to the Global Burden of Diseases, Injuries, and Risk Factors Study (GBD) 2019, noncommunicable diseases (NCDs) such as cardiovascular diseases (CVDs) are leading causes of disability-adjusted life years (DALYs) among people aged 50 years or over [4]. Stroke and ischemic heart disease were reported as the top two leading causes of DALYs in all ages, and in the 50–69 age group in China during 2019 [5]. Self-rated health (SRH), which is a widely-used tool for the measurement of health status, can effectively detect suboptimal health status (SHS) [6]; poor SRH is associated with higher future risk of chronic diseases, multimorbidity, and mortality [6,7,8,9,10,11]. The reliability of SRH has been assessed in different populations [9,12,13]. Generally, moderate to substantial test-retest reliability of SRH was reported among adults [9,12,13].

Labour-market characteristics refers to a wide range of work-related conditions. Previous studies provided evidence for associations between some of the labour-market characteristics and poor SRH. For example, studies of Korean workers indicated that precarious employment, such as temporary work, was associated with 1.31 times higher chance of poorer SRH for men and 1.34 times higher chance of poorer SRH for women [14]. The effects of long working hours on health is well studied in developed countries [15], although there is no strict definition of long working hours and the cut-offs for defining long working hours (e.g., ≥55 h/week) differ across studies [16,17,18,19,20]. Working long hours is associated with increased risk of chronic diseases and poor SRH [17,19,20,21,22,23]. Two of the possible connections linking long working hours and poor health outcomes are physiological and behavioural mechanisms [16]. Other labour-market characteristics, such as unemployment [24], occupation [25], and lack of paid sick leave, are also associated with poor SRH [26]. However, existing evidence mainly comes from developed and western countries, and studies on associations between labour-market characteristics and SRH among the Chinese working population are scarce [16]. The current study, for the first time, used nationally-representative data from China to investigate associations between fourteen labour-market characteristics and SRH in the Chinese middle-aged and older working population. As previous studies of China reported gender and regional differences in SRH, and the effects of occupational stress [27,28,29], we also tested the interactions between sexes and among regions for each labour-market characteristic.

## 2. Materials and Methods

We used data from the China Health and Retirement Longitudinal Study (CHARLS), a nationally-representative study conducted among community residents aged 45 years or older living in 28 Chinese provinces [30,31]. Institutionalised individuals were not sampled [30]. CHARLS is intended to provide micro, longitudinal data, covering health measures and indicators of socio-economic status, of the middle-aged and older working population, for the study of Chinese age-related demographic issues [30]. Information on family, health status, healthcare, work circumstances, and economic status was collected [31]. The CHARLS national baseline survey was conducted between 2011 and 2012 [31], and four national waves are available. The first national survey was used in the present study. Further information about CHARLS can be found in the CHARLS cohort profile and users’ guide [30,31]. The sample-selection procedure used for the current study is illustrated in Figure 1.

### 2.1. Labour-Market Characteristics for Working Population

For the purposes of this paper, working population refers to respondents who were holding, at the time of the survey, at least one non-agricultural job or who did non-agricultural work for at least one hour during the previous week. This includes people who were employed (earning a wage), self-employed, working unpaid in a family business, temporarily laid-off, on leave, or doing on-job training but planning to return within six months. Respondents doing only housework or unpaid activities such as voluntary work were not included in these analyses.

Labour-market characteristics in the working population include employment status, weekly working hours, work sector (public or private), quintiles of earned income (quintiles based on income of CHARLS participants), occupation, length of current employment, and whether workers had any social-support insurance (including pension, health insurance, unemployment insurance, worker’s injury insurance, and maternity insurance).

Participants without clear employment status (being either employed or self-employed, or doing unpaid work in a family business) were put into a separate category (Not enough information to classify). Weekly working hours included hours spent on the main job plus additional jobs. The public sector includes government organisations, institutions operated as government units, non-government organisations, and firms that were 100% state owned, state-controlled, 100% collective-owned, or collective-controlled. Earned income refers to after-tax salary, plus all bonuses and subsidies for the employee, and income or wages from additional jobs. For self-employed workers and unpaid workers in family business, earned income meant estimated net income for the previous year. For this group, income information was only requested in the household section; therefore, the income could not be considered as individual-earned income. Occupation categories were based on self-reported job descriptions, coded using International Standard Classification of Occupations (ISCO) in CHARLS [30]. Major groups of ISCO-08 consist of 1: Managers; 2: Professionals; 3: Technicians and associate professionals; 4: Clerical support workers; 5: Service and sales workers; 6: Skilled agricultural, forestry, and fisheries workers; 7: Craft and related trades workers; 8: Plant and machine operators, and assemblers; 9: Elementary occupations; and 10: Armed forces occupations [32]. According to skill level, occupations in the current study were categorised into Managers (except hospitality, retail, and other services managers) and professionals, Technicians and associate professionals, Clerks and workers, and Elementary occupations. In addition, government employees (such as civil servants and formal employees of an establishment) and formal employees of an establishment in institutions were not asked about employer-provided insurance in CHARLS because these people were considered to have all necessary insurance by default. Therefore, those people were categorized to “At least one type of insurance” in the section about employer-provided/self-employed covered insurance.

### 2.2. Labour-Market Characteristics for Employees

In addition to characteristics available to the full analytical sample, there are additional characteristics available for “employees”, including government employees and company employees. In the work section of the CHARLS questionnaire, some questions were only asked of employees. These work characteristics include whether a person was in a supervisory position, type of employing entity, employment type at current workplace, whether a person had a written labour contract for their current workplace/labour-dispatch company, labour-contract period, number of days for paid vacation or paid sick leave, and fringe benefits.

Team leaders, government officials (including section chiefs, directors of a division, and directors-general of bureaus, and above), village leaders, township leaders, division managers, and general managers were not asked the question “Are you in a position to supervise others”, however, they were still considered to be supervisors in the current analysis. In addition, government officials were not asked in CHARLS about their employment type at current workplace, and they were considered to be contract workers.

### 2.3. Self-Rated Health

SRH was determined by the question “Would you say your health is very good, good, fair, poor or very poor?”. In the current study, SRH was dichotomised into good or fair (very good, good, and fair) and poor (poor and very poor).

### 2.4. Covariates

Sociodemographic factors included age, sex, marital status (unmarried including separated, divorced, widowed, and never married), education level, residence type, and region. Whether the administrative village/neighbourhood that each household belongs to was in an urban or rural area was defined by the National Bureau of Statistics (NBS) of China [30]. Considering the large number of internal migrant workers and the urban-rural gap in China, residence type was defined based on their official household registration (known as Hukou) status, including: urban (urban residents with non-agricultural Hukou), migrant (rural residents with non-agricultural Hukou or urban residents with agricultural Hukou), and rural (rural residents with agricultural Hukou) [33,34], According to the division of NBS in 2011, the 28 provinces included in CHARLS were divided into four regions (East, Central, West, and Northeast) based on their social- and economic-development levels [35]. Health behaviours include smoking and alcohol-consumption status. Occasional drinker was defined as a person who drinks alcohol once a month or less, and frequent drinker meant a person who drinks more than once a month.

### 2.5. Statistical Analyses

We calculated the prevalence of poor-SRH among the working population according to sample characteristics and labour-market characteristics. The chi-squared test was used to investigate association between each sample characteristic and SRH. Due to the design of the CHARLS questionnaire, some work-related questions were only asked of paid employees. As a result, there are two groups of labour-market characteristics in the current study. One group is available for the whole sample, and the other is only available for paid employees. Thus, we analysed the two groups separately. To investigate the association between labour-market characteristics and SRH, multiple logistic regression models of the effect of each labour-market characteristic were estimated, controlling for sociodemographic factors (age, sex, marital status, education, residence type, region) and health behaviours (smoking and alcohol-consumption status). Covariates were added to the model sequentially. 

Interactions between sex and each labour-market characteristic, and between region and each labour-market characteristic were tested, as sex and region are potentially important effect modifiers. Although there was some evidence for the interactions between sex and employer-provided insurance (*p* = 0.05), and between region and weekly working hours (*p* = 0.04), generally little evidence was found for statistical heterogeneity. Therefore, homogeneity was assumed, and combined results were reported (Region-specific results was presented in Appendix A). Furthermore, data from participants who were not asked certain questions were considered missing (e.g., labour-contract period was only asked among those having labour contracts). Data were missing for labour-market characteristics for various reasons (such as implausible values, questions not answered, and questions not asked). Such data were combined and analysed as a separate category. All analyses (including Appendix A) were performed using Stata/MP 16.1 [36].

## 3. Results

### Sample Characteristics

Descriptive characteristics of the analytical sample are shown in Table 1. (For Table 1, Table 2 and Table 3, percentages may not total 100% due to rounding.) The distribution of SRH measured using the five-point scale among the whole sample is shown in Appendix A. The overall prevalence of poor or very poor SRH in subjects was 14%. Older age, being female, unmarried, of lower education level, rural residence type, living in Eastern or Central regions, and having a higher drinking frequency, were associated with higher prevalence of poor SRH. Older age groups had significantly higher prevalence of poor SRH (*p* for trend < 0.001). Migrant status was not significantly associated with higher prevalence of poor SRH, although the prevalence of poor SRH among migrant residents (13%) was slightly higher than among their urban counterparts (11%). There was a significant difference between urban and rural residents (*p* < 0.001). The prevalence of poor SRH is significantly higher in Central (*p* = 0.01) and Western China (*p* < 0.001), with the highest prevalence of 18% occurring in Western China, compared with 12% in Eastern China.

Table 2 shows the prevalence and odds ratios (ORs) of poor SRH for each labour- market characteristic in the working population. When controlled for age and sex, people who are self-employed and doing unpaid work in a family business have 1.23 times and 2.34 times chance, respectively, of poor SRH compared with employed participants. After further adjustment for marital status, education level, residence type, living regions, smoking status, and alcohol consumption, the ORs dropped slightly but still indicate greater chance of poor SRH for self-employed and unpaid-family-business groups.

Only 21% of respondents worked standard hours (40–49 h/week), while 30% of respondents worked ≥60 h/week. In a fully adjusted model, compared with people working standard hours, working for fewer hours (1–39 h/week) and long hours (≥60 h/week) are both associated with higher odds of poor SRH (OR = 1.60 and 1.24). Working in the private sector is associated with a 1.24 times greater chance of poor SRH, compared with working in the public sector. In addition, results suggest a strong dose-response relationship between income level and poor SRH (*p* for trend < 0.001).

Compared with managers/professionals, technicians/associate professionals had lower odds of poor SRH (fully adjusted OR = 0.65). Clerks/workers and elementary occupations had odds of poor SRH similar to managers/professionals (fully adjusted OR = 0.95 and 0.96). Fifty-three percent of respondents had worked in their current job for 10 years or less. Compared with those respondents, people working in their current job for over 10 years had lower odds (OR = 0.91) of poor SRH, controlling for age and sex. While the effects of occupation and length of current employment on SRH were not statistically significant, after adjustment of age and sex, the ORs still indicate the direction of those associations. Finally, compared with participants having at least one employer-provided form of insurance, participants without any employer-provided insurance or self- covered insurance had higher odds of poor SRH (OR = 1.35), although this difference disappeared in the fully-adjusted model (OR = 1.06).

Table 3 shows the prevalence and ORs of poor SRH for each labour-market characteristic among employees. People who were not in supervisory positions had 1.86 times greater odds of poor SRH, controlling for age and sex. The association remained significant after further adjustment (OR = 1.62). Only 8% of employees received wages from labour-dispatch companies, and they had 1.27 times greater odds of poor SRH compared with those receiving wages from place of work, controlling for age and sex. Similarly, casual/part-time workers had 1.34 times greater chance of poor SRH, compared with contract workers, controlling for age and sex. However, as reported in Table 3, there was only weak evidence for the effect of receiving wages from labour-dispatch companies and causal/part-time work on SRH.

In addition, compared with the 24% of employees who had written labour contracts, employees without such contracts had 1.41 times higher odds of poor SRH, controlling for age and sex. Compared with employees who had paid vacation/sick leave, those without any paid vacation/sick leave had 1.45 (age- and sex-adjusted) times greater odds of poor SRH. Similarly, compared with employees with fringe benefits, employees without any fringe benefits had 1.27 (age- and sex-adjusted) times higher odds of poor SRH. As shown in Table 3, further adjustment reduced these odds ratios, but while non-significant, the direction of the associations remained the same.

## 4. Discussion

### 4.1. Main Finding of This Study

In this study, we used national representative data to assess the associations between a wide range of labour-market characteristics and SRH among the Chinese middle-aged and older working population. Several labour-market characteristics were associated with poor SRH among Chinese middle-aged and older workers, including doing unpaid work in family businesses, working for less than 40 h/week or working for 60 h/week or over, working in private sectors, earning lower income, being without any employer-provided or self-provided insurance, not being in a supervisory position, and being without a written labour contract. The associations were generally statistically significant when controlled for age and sex, although less so when fully adjusted. Residence type and region appear to have acted as confounding variables in several associations between labour-market characteristics and SRH.

### 4.2. Findings in the Context of Existing Studies

The current study used a widely used version of SRH scale (WHO version) as the measurement of outcome [37]. The distribution of SRH in the current study is shown in Appendix A. Among 3864 working individuals, 34.03% reported “very good” or “good” SRH, 51.71% reported “fair” SRH, and 14.26% reported “poor” or “very poor” SRH. Compared with a previous study based on 12431 CHARLS wave-1 participants (“very good”/“good”: 23.4%, “fair”: 51.2%, “poor”/“very poor”: 25.5%), SRH reported by the working population in the current study was generally better [9]. This may be due to the “healthy worker survivor effect”, as employed individuals tend to be healthier than the general population which includes those who left employment [38]. Previous research from western countries also reported the distribution of the WHO version SRH among middle-aged and older adults. In the study of Jürges et al. using the Survey of Health, Ageing and Retirement in Europe (SHARE), among 11643 participants (aged 50 years and over), 60.5% of them reported “very good” or “good” SRH, 29.8% reported “fair” SRH, and 9.7% reported “poor” or “very poor” SRH [39]. Compared with western populations, Chinese population are more likely to report a “fair” SRH.

Some of the labour-market characteristics used in this study generally reflect social disadvantage, for example: working for very long hours, earning lower income, being without any employer-provided insurance or without any formal labour contract. Those characteristics may simply indicate that the individuals have lower socioeconomic status (SES), which is usually associated with poorer health [40].

In addition, some country-specific factors, such as migrant status, should be considered when interpreting the results of the current analysis.

Due to the unbalanced regional development and urban-rural gap, there is a huge number of rural migrant workers in China. Rural migrant worker here refers to workers holding rural household registration but who engage in non-agricultural work in their home area, or work outside their home area for six months or more per year [34]. According to the NBS, there were 285.6 million rural migrant workers in 2020, accounting for 38.05% of the total employed population [34]. Fifty-three percent of rural migrant workers worked in the Eastern region where economic development was more advanced [34]. Compared with their urban counterparts, rural migrant workers have many disadvantages, including less access to urban services and lower job stability [41,42]. In addition, rural migrant workers also have higher rates of health-risk behaviours, such as alcohol use and smoking [43], and a poorer mental health status [44,45].

The internal rural migration of workers in China is economically driven and usually temporary. In accordance with previous studies, there was no significant difference in the health status between urban residents and the migrant population [46,47], while compared with both the urban and migrant populations, rural residents reported worse health status. Moreover, the population that returns after migration experience had worse physical health compared with the migrant population, as unhealthy migrants usually choose to return to their hometowns [47]. Over half of rural migrant-working individuals worked in the Eastern region in 2020, where the economy is more developed. Appendix A shows the four major economic regions in China. In Appendix A, we report the results of associations between labour-market characteristics and SRH in working population according to region. Generally, the biggest difference is between the Northeast and other regions. However, it should be noted that the number of participants from the Northeast only accounts for 7.6% of the whole sample.

The current study found that the prevalence of poor SRH is significantly higher among women, which is in consistent with previous studies [48]. In China, women usually take more responsibility for the care of the family. A recent study of Chinese females found that the double burden of work and informal care significantly increased the probability of reporting poor SRH (2.35% higher probability compared with women with no care burden), by reducing exercise time and increasing psychological stress [48].

As the CHARLS baseline wave did not ask about unemployment, we could not unequivocally define this important variable. However, an earlier study conducted in three North-Western Chinese cities reported that, compared with the employed population, unemployed people had 1.40 times greater chance of poorer SRH (95% CI, 1.25–1.55) [49]. In addition, consistent with previous studies, our results indicate that both short working hours (1–39 h/week) and long working hours (≥60 h/week) are associated with an increased risk of poor SRH [18]. The adverse effect of long working hours on health status has been supported by many previous studies [15,21,22]. However, for those working short hours, the higher odds of poor SRH might be due to reverse causality, as people with worse health status may reduce their working hours.

### 4.3. Strengths and Limitations

This study used nationally-representative data to assess the effect of a wide range of labour-market characteristics and SRH among the Chinese working population, and investigated the importance of migrant status and region in these associations. Our study provides the basic evidence of understanding labour-market characteristics and the health conditions of the Chinese middle-aged and older workforce. In addition, the present study used information from people who are currently working, instead of employment history, thus the recall bias was greatly reduced.

However, this study also has several limitations. First, although we only included participants who are currently holding at least one non-agricultural job, still 44.3% of participants reported engagement in agricultural work for over 10 days in the last year (86.2% engaged in their own household agricultural work and 61.1% were rural residents). One possible reason for the large proportion of participants doing non-agricultural work but engaged in agricultural work for over 10 days at the same time, is that migrant workers (who account for 29% of the analytical sample) and rural residents (who account for 45% of the analytical sample) may help with their household agricultural work during busy farming seasons. As hours of agricultural work were not counted in the current analysis, this may have affected the association between weekly working hours and poor SRH. Second, as the outcome of interest is SRH, there could be a potential measurement bias. Third, the data we used were from the national baseline wave of CHARLS, which was conducted between 2011 and 2012, therefore requiring future studies assessing the longitudinal association between labour-market characteristics and health outcomes in the Chinese middle-aged and older workforce. Fourth, we decided to keep a specific category for missing values in employment variables. It would be possible to use an alternative approach, such as multiple imputations, but we believe that the proportion missing due to unanswered questions about employment characteristics was relatively low, so multiple imputation would possibly only have little impact on our findings. Fifth, although we used a wide range of working characteristics, some other work-related characteristics, such as support and training provided by employers, or experience of workplace violence and/or bullying, were not available. Finally, we cannot exclude the potential reverse causality, whereby people move to different occupation because of their ill health.

## 5. Conclusions

Several labour-market characteristics were associated with higher odds of poor SRH among Chinese middle-aged and older workers when controlled for age and sex. Those characteristics included doing unpaid work in family businesses (OR = 2.34), working for less than 40 h/week (OR = 1.83) or for 60 h/week or more (OR = 1.39), working in private sectors (OR = 1.45), earning lower income (4th quintile: OR = 2.13; 5th quintile: OR = 3.19), being without any employer-provided or self-provided insurance (OR = 1.35), not being in a supervisory position (OR = 1.86), and being without any written labour contract (OR = 1.41). Working subjects with the above characteristics, especially those doing unpaid work in family businesses or who earn lower incomes, are likely to be more socially disadvantaged and have a worse general health status. Most of the adverse labour-market conditions examined in this study can be improved. Based on our results, there are two main aspects needing improvement. The first is to improve and firmly implement labour laws regarding discrimination (e.g., gender discrimination, age discrimination), long working hours, wages and allowances, and labour contracts. The second aspect is to improve the social insurance system regarding employer-provided insurance and insurance for individuals doing unpaid work in family businesses.

## Figures and Tables

**Figure 1 ijerph-20-04748-f001:**
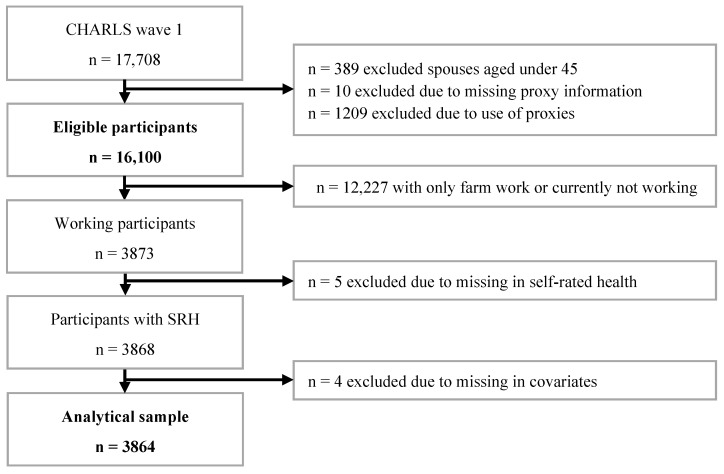
Flowchart of the sample-selection procedure.

**Table 1 ijerph-20-04748-t001:** Sample characteristics and self-rated health in the Chinese working population.

	N (Col %)	Self-Rated Health (Row %)		*p*-Values ^1^
		Very Good/Good/Fair	Poor/Very Poor	
	N = 3864	N = 3313 (86)	N = 551 (14)	
**Age**				
45–49	1424 (37)	1271 (89)	153 (11)	Ref
50–54	826 (21)	711 (86)	115 (14)	0.02
55–59	890 (23)	746 (84)	144 (16)	0.001
60 or over	724 (19)	585 (81)	139 (19)	<0.001
**Sex**				
Male	2456 (64)	2150 (88)	306 (13)	Ref
Female	1408 (36)	1163 (83)	245 (17)	<0.001
**Marital status**				
Married	3603 (93)	3104 (86)	499 (14)	Ref
Unmarried	261 (7)	209 (80)	52 (20)	0.01
**Education**				
High school or above	924 (24)	825 (89)	99 (11)	Ref
Middle school	1112 (29)	983 (88)	129 (12)	0.53
Elementary school	790 (20)	680 (86)	110 (14)	0.04
Lower than elementary school	539 (14)	445 (83)	94 (17)	<0.001
Illiterate	499 (13)	380 (76)	119 (24)	<0.001
**Residence**				
Urban	1042 (27)	926 (89)	116 (11)	Ref
Migrant	1103 (29)	961 (87)	142 (13)	0.22
Rural	1719 (45)	1426 (83)	293 (17)	<0.001
**Regions**				
East	1613 (42)	1422 (88)	191 (12)	Ref
Central	1056 (27)	896 (85)	160 (15)	0.01
West	903 (23)	743 (82)	160 (18)	<0.001
Northeast	292 (8)	252 (86)	40 (14)	0.37
**Smoking status**				
Never smoker	2008 (52)	1711 (85)	297 (15)	Ref
Former smoker	347 (9)	297 (86)	50 (14)	0.85
Current smoker	1509 (39)	1305 (87)	204 (14)	0.29
**Alcohol drinking**				
Not at all	2088 (54)	1715 (82)	373 (18)	Ref
Occasional drinker	395 (10)	349 (88)	46 (12)	0.003
Frequent drinker	1381 (36)	1249 (90)	132 (10)	<0.001

^1^*p* values for difference in prevalence of poor/very poor self-rated health.

**Table 2 ijerph-20-04748-t002:** Association between each labour-market characteristic and poor self-rated health in the Chinese working population.

	N (Col %)	Poor SRH (Prevalence %)	OR (95% CI) Age- and Sex-Adjusted	OR (95% CI) Fully Adjusted ^1^
	N = 3864	N = 551 (14)		
**Employment status**				
Employed	2449 (63)	293 (12)	Ref	Ref
Self-employed	1055 (27)	154 (15)	1.23 (1.00, 1.52)	1.15 (0.93, 1.43)
Unpaid family business	259 (7)	71 (27)	2.34 (1.72, 3.18) **	2.07 (1.51, 2.84) **
Not enough information to classify	101 (3)	33 (33)	3.08 (1.99, 4.79) **	2.55 (1.62, 4.02) **
**Weekly working hours**				
1–39 h/week	735 (19)	134 (18)	1.83 (1.35, 2.48) **	1.60 (1.17, 2.18) *
40–49 h/week	794 (21)	78 (10)	Ref	Ref
50–59 h/week	731 (19)	76 (10)	1.01 (0.73, 1.42)	0.92 (0.65,1.30)
≥60 h/week	1148 (30)	158 (14)	1.39 (1.04, 1.85) *	1.24 (0.92, 1.67)
Missing ^2^	456 (12)	105 (23)	2.35 (1.69, 3.25) **	1.98 (1.41, 2.77) **
**Public or private sectors**				
Public	846 (22)	88 (10)	Ref	Ref
Private	3004 (78)	461 (15)	1.45 (1.13, 1.85) *	1.24 (0.94,1.64)
Missing ^3^	14 (0.4)	2 (14)	1.27 (0.27, 5.83)	0.95 (0.20, 4.52)
**Earned income**				
1 (Highest)	626 (16)	42 (7)	Ref	Ref
2	643 (17)	67 (10)	1.60 (1.07, 2.39) *	1.59 (1.06, 2.40) *
3	613 (16)	55 (9)	1.25 (0.82, 1.90)	1.18 (0.77, 1.81)
4	651 (17)	95 (15)	2.13 (1.45, 3.13) **	1.92 (1.29, 2.86) *
5 (Lowest)	640 (17)	139 (22)	3.19 (2.19, 4.65) **	2.72 (1.83, 4.02) **
Missing ^3^	691 (18)	153 (22)	3.42 (2.37, 4.94) **	2.99 (2.04, 4.38) **
**Occupation**				
Managers and professionals	942 (24)	122 (13)	Ref	Ref
Technicians and associate professionals	118 (3)	9 (8)	0.57 (0.28, 1.16)	0.65 (0.32, 1.34)
Clerks and workers	1996 (52)	262 (13)	1.01 (0.80, 1.27)	0.95 (0.74, 1.21)
Elementary occupations	423 (11)	61 (14)	1.06 (0.76, 1.48)	0.96 (0.68, 1.36)
Missing ^2^	385 (10)	97 (25)	1.97 (1.45, 2.67) **	1.66 (1.21, 2.28) *
**Length of current job**				
≤10 years	2057 (53)	281 (14)	Ref	Ref
>10 years	1419 (37)	172 (12)	0.91 (0.74, 1.12)	1.00 (0.81, 1.24)
Missing ^3^	388 (10)	98 (25)	1.89 (1.45, 2.46) **	1.77 (1.35, 2.33) **
**Employer-provided/self-employed covered insurances**				
At least one insurance	887 (23)	91 (10)	Ref	Ref
None	2516 (65)	372 (15)	1.35 (1.05, 1.73) *	1.06 (0.80, 1.40)
Missing ^3^	461 (12)	88 (19)	1.55 (1.11, 2.16) *	1.31 (0.93, 1.84)

Abbreviations: *SRH* self-rated health, *OR* odds ratio, *CI* confidence interval. ^1^ Adjusted for age, sex, marital status, education, residence, region, smoking status, and alcohol consumption. ^2^ Implausible value or not answered. ^3^ Implausible values, not answered, or not asked. * *p* < 0.05. ** *p* < 0.001.

**Table 3 ijerph-20-04748-t003:** Association between each labour-market characteristic and poor self-rated health among Chinese employees.

	N (Col %)	Poor SRH (Prevalence %)	OR (95% CI) Age- and Sex-Adjusted	OR (95% CI) Fully Adjusted ^1^
	N = 2449	N = 293 (12)		
**Supervisory position**				
Yes	340 (14)	23 (7)	Ref	Ref
No	2052 (84)	262 (13)	1.86 (1.19, 2.91) *	1.62 (1.01, 2.58) *
Missing ^2^	57 (2)	8 (14)	1.82 (0.76, 4.35)	1.45 (0.59, 3.55)
**Receiving wages from**				
Place of work	2223 (91)	258 (12)	Ref	Ref
Labour dispatch company	202 (8)	29 (14)	1.27 (0.84, 1.93)	1.11 (0.72, 1.70)
Missing ^2^	24 (1)	6 (25)	2.13 (0.83, 5.47)	1.70 (0.65, 4.47)
**Employment type at current workplace**				
Contract worker	753 (31)	72 (10)	Ref	Ref
Labour dispatch worker	182 (7)	19 (10)	1.08 (0.63, 1.84)	0.99 (0.57, 1.71)
Casual/Part-time worker	1299 (53)	174 (13)	1.34 (1.00, 1.80)	1.14 (0.81, 1.60)
Missing ^3^	215 (9)	28 (13)	1.33 (0.83, 2.12)	1.26 (0.78, 2.03)
**Labour contract in written form**				
Yes	588 (24)	51 (9)	Ref	Ref
No	1787 (73)	226 (13)	1.41 (1.02, 1.95) *	1.21 (0.84, 1.73)
Missing ^2^	74 (3)	16 (22)	2.53 (1.34, 4.76) *	2.09 (1.08, 4.04) *
*Asked if had labour contract:* **Labour contract period**				
Defined period	284 (12)	26 (9)	Ref	Ref
Not defined	299 (12)	25 (8)	0.88 (0.50, 1.57)	0.90 (0.50, 1.62)
Same as the term of the project	7 (0.3)	0 (0)	-	-
Missing ^3^	1859 (76)	242 (13)	1.35 (0.88, 2.08)	1.17 (0.75, 1.84)
**Paid vacation/paid sick leave** *(Num. of Days)*				
≥1	306 (13)	25 (8)	Ref	Ref
0	2083 (85)	258 (12)	1.45 (0.94, 2.23)	1.21 (0.76, 1.92)
Missing ^3^	60 (2)	10 (17)	1.82 (0.81, 4.06)	1.39 (0.61, 3.19)
**Fringe benefits**				
Yes	722 (30)	71 (10)	Ref	Ref
No	1650 (67)	206 (13)	1.27 (0.95, 1.69)	1.22 (0.91, 1.63)
Missing ^2^	77 (3)	16 (21)	2.10 (1.14, 3.86) *	1.86 (1.00, 3.46)

Abbreviations: *SRH* self-rated health, *OR* odds ratio, *CI* confidence interval. ^1^ Adjusted for age, sex, marital status, education, residence, regions, smoking status, and alcohol consumption. ^2^ Implausible value or not answered. ^3^ Implausible values, not answered, or not asked. * *p* < 0.05.

## Data Availability

The data that support the findings of this study are openly available on CHARLS project website at http://charls.pku.edu.cn/ (accessed on 1 January 2021).

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
