# Peer review of "Labour-Market Characteristics and Self-Rated Health: Evidence from the China Health and Retirement Longitudinal Study"

_ijerph, 2023, doi:10.3390/ijerph20064748_

Round 1

Reviewer 1 Report

Thank you for the opportunity to review this manuscript. This population-based study explores Labour Market Characteristics and Self-rated Health: Evidence 2 from China Health and Retirement Longitudinal Study. The methodology of the study is described in detail. Overall, the article is well written.  I have only two suggestions for further improvement of the manuscript, as detailed below:

1. Consider to

A. Present (in the results section) the distribution of the SRH variable for the entire population (not only as a binary variable);

B. compare (in a discussion section) SRH distribution in the current study to other findings from Chines and worldwide populations 

2. lines 190-199 present ODs with a nonsignificant effect. It should be mentioned in the text.

Author Response

Overall, the article is well written. I have only two suggestions for further improvement of the manuscript, as detailed below:

Consider to

  1. Present (in the results section) the distribution of the SRH variable for the entire population (not only as a binary variable);

We have added Table A1 to the appendix. As there are already 3 large tables in the main text we would prefer to put this information in the appendix (although we would be happy to move it to the main text if Editors or Reviewer prefer so).

  1. compare (in a discussion section) SRH distribution in the current study to other findings from China and worldwide populations

We agree that such a comparison is helpful. We added such comparison to the first paragraph of section 4.2.

Lines 190-199 present ORs with a nonsignificant effect. It should be mentioned in the text.

We have made such changes in the text.

Reviewer 2 Report

Study summary: This manuscript investigates the impact of labour market characteristics, migrant status and region on self-rated health in Chinese middle-aged and older population. For the analyses, nationally representative data from the China Health and Retirement Longitudinal Study (CHARLS) was used. The study findings show that middle-aged and older adults who do unpaid 234 family business, work for less than 40 h/week or work for 60 h/week or over, work in private sectors, earn lower income, have no employer-provided or self-employed covered insurance, have no supervisory position, and have no written-form labour contract experience more likely poor self-rated health. Furthermore, residence type and region were significant confounder in the associations between self-rated health and some labor market characteristics.

This study provides valuable and innovative findings based on the data from a nationally representative study. Nevertheless, there are some issue that should be improved:

Introduction

Line 28: Abbreviation “CVD” should be written out first. I would also expect references for the statement within the lines 27-28.

Lines 41-49: The state of research on associations between labour market characteristics and self-rated health should be elaborate in more detail, although the most previous studies were conducted outside of China. There is also no clear argumentation why particularly the relationship between labour market characteristics and self-rated health should be analysed. It should be also explained why the role of migrant status and region has been investigated.

You stated in lines 131-132, that employees and employer were analysed separately. In the introduction, the explanation including research evidence should be added why this two groups should be observed separately.

Line 138: You have analysed the interactions between sex and each labour market characteristic and between region and each market characteristic. In the introduction, explanation including research evidence should be added why the analysing of these interactions might be relevant.

Materials and Methods

Paragraph within lines 51-57: More information about the study CHARLS should be given here, such as how the data were collected and from who (for example target person and all household members), in which cases data from proxy person was collected, how the representativity of the study sample was ensured etc.). Additionally, the reason why the data provided from proxies (n=1,209) and those who were working on a farm were excluded from the analyses is missing.

Line 53: The term “elderlies” should be replaced for example with „older adults“. „Elderly“ is supposed to be a ageist term (Avers et al. 2011).

Line 65: The “only” should be removed from the sentence “Respondents doing only agricultural work, housework…” as it may implicate the subjective evaluation of the authors regarding these types of employment status.

Results

Line 170: In Table 2, the reference category of the employment status is “employed”. Thus, in this sentence, the comparison is made with “employed participants” instead of "employees". Please, check and correct this sentence if needed.

Nearly in all labour market variables of Table 2 and some of Table 3, “missing” category is statistically significant. These results should be also reported in the result section as well as discussed in the discussion section.

Lines 190-192: According to Table 2, there is no statistically significant association between occupation and SRH (except for category “missing”) in both models. This should be noted in this part of text.  

Lines 193-196: According to Table 2, there is no association between “working in current job for over 10 years and SRH by controlling for age and sex”. Therefore, please check and correct this part if needed.

Lines 212-216: The reported associations are not statistically significant. This should be noted in the text.

Lines 218-220: This association was also not statistically significant. This should be reported.

Lines 223-226: The same issue as previously mentioned regarding the lines 212-220.

Table 3: In the table or under the table, it should be noted that the variable Labour contract period only the person are included who have a written labour contract.

Overall: One of the goals of the study was to examine the role of migrant status and of region in the association between labour market characteristics and SRH. However, these associations are not reported in the results section nor in the tables. These results should be described in the results section as well as in the additional table or tables (for example in Appendix).

References

Avers, Dale; Brown, Marybeth; Chui, Kevin K.; Wong, Rita A.; Lusardi, Michelle (2011): Editor's message: Use of the term "elderly". In: Journal of geriatric physical therapy (2001) 34 (4), S. 153–154. DOI: 10.1519/JPT.0b013e31823ab7ec.

Author Response

This study provides valuable and innovative findings based on the data from a nationally representative study. Nevertheless, there are some issue that should be improved:

Introduction

Line 28: Abbreviation “CVD” should be written out first.

Changes have been made accordingly.

I would also expect references for the statement within the lines 27-28.

The sentence is now rephrased and reference is added.

Lines 41-49: The state of research on associations between labour market characteristics and self-rated health should be elaborate in more detail, although the most previous studies were conducted outside of China.

More information on previous research has been added in the Introduction in reference to this comment.

There is also no clear argumentation why particularly the relationship between labour market characteristics and self-rated health should be analysed.

We have now added such argument into the first paragraph of the Introduction.

It should be also explained why the role of migrant status and region has been investigated.

The discussion on the role of internal migration and migrant status as well as the role of region variable is included in the Discussion (lines 313-336).

You stated in lines 131-132, that employees and employer were analysed separately. In the introduction, the explanation including research evidence should be added why this two groups should be observed separately.

In lines specified above we explained that two groups of labour market characteristics were available in the dataset, one for the full sample and the second only for paid employees. The analysis was divided into these two parts entirely because of the design of the CHARLS questionnaire, rather than of previous evidence. In other words, paid employees and the whole working population (not employers) were analysed in two separate analyses because some work-related questions (such as those related to fringe benefits) were only asked among paid-employees. These questions are described in detail in Table 3 (Association between each labour market characteristic and poor self-rated health among employees). We hope that the explanation (lines 129-131 in original manuscript) and this justification make the reasons clearer.

Line 138: You have analysed the interactions between sex and each labour market characteristic and between region and each market characteristic. In the introduction, explanation including research evidence should be added why the analysing of these interactions might be relevant.

Relevant evidence was added in the introduction.

However, the vast majority of existing evidence on gender differences in the role of work environment on health mainly comes from European studies. Since effect modification by gender is plausible and evidence from China is missing, we assessed these potential effect modifiers in our analyses.

Materials and Methods

Paragraph within lines 51-57: More information about the study CHARLS should be given here, such as how the data were collected and from who (for example target person and all household members), in which cases data from proxy person was collected, how the representativity of the study sample was ensured etc.).

Additionally, the reason why the data provided from proxies (n=1,209) and those who were working on a farm were excluded from the analyses is missing.

Thank you. We were probably too cautious about the lengths of the manuscript. We have now added more information on CHARLS study in section 2 “Materials and methods”.

We decided to exclude self-rated health (SRH) reported by proxies after careful consideration, as we believe SRH is supposed to be self-reported and subjective; SRH reported by proxies of respondents is likely to be affected by subjective views of these proxies, that then of the study subject, and as such may not be comparable to the SRH reported directly by respondents.

The current study focused on Chinese non-agricultural workers. The main reason is that the study did not collect many labour market characteristics that can be used for farm workers in China (for example, working in public or private sectors, employer-provided insurances, supervisory position, labour contract, labour contract period, paid vacation or paid sick leave, and fringe benefits are characteristics that have no or very limited meaning for Chinese farm workers).

Line 53: The term “elderlies” should be replaced for example with „older adults“. „Elderly“ is supposed to be a ageist term (Avers et al. 2011).

References:  Avers, Dale; Brown, Marybeth; Chui, Kevin K.; Wong, Rita A.; Lusardi, Michelle (2011): Editor's message: Use of the term "elderly". In: Journal of geriatric physical therapy (2001)34 (4), S. 153–154. DOI:10.1519/JPT.0b013e31823ab7ec.

Response: Changes have been made accordingly.

Line 65: The “only” should be removed from the sentence “Respondents doing only agricultural work, housework…” as it may implicate the subjective evaluation of the authors regarding these types of employment status.

Thank you and sorry if the use of language caused misunderstanding. The sentence has been rephrased.

Results

Line 170: In Table 2, the reference category of the employment status is “employed”. Thus, in this sentence, the comparison is made with “employed participants” instead of "employees". Please, check and correct this sentence if needed.

Thanks, done.

Nearly in all labour market variables of Table 2 and some of Table 3,“missing” category is statistically significant. These results should be also reported in the result section as well as discussed in the discussion section.

These results are now reported and Discussion on missing data has been added in section 4.3.

Lines 190-192: According to Table 2, there is no statistically significant association between occupation and SRH (except for category “missing”)in both models. This should be noted in this part of text.

Changes have been made accordingly. Thank you for this suggestion.

Lines 193-196: According to Table 2, there is no association between “working in current job for over 10 years and SRH by controlling for age and sex”. Therefore, please check and correct this part if needed.

According to table 2, “people working in current job for over 10 years had lower odds (OR=0.91) of poor SRH, controlling for age and sex”. We agree that this association is not statistically significant (95% CI, 0.74-1.12) and it should be clearly expressed. Therefore, changes have been made accordingly.

Lines 212-216: The reported associations are not statistically significant. This should be noted in the text.

Lines 212-213: The association between supervisory position and SRH remained marginally significant after further adjustment (OR=1.62, 95% CI: 1.01-2.58, p<0.05). However, changes were made  for this description.

Lines 214-216: we agree and changes have been made accordingly.

Lines 218-220: This association was also not statistically significant. This should be reported.

Line 218: “employees without such contract had 1.41 times higher odds of poor SRH, controlling for age and sex”, no changes were made for this description, as the 95% CI is 1.02-1.95 and p-value is less than 0.05. It might be borderline significant but not non-significant. Lines 219-220: “among those employees with labour contract, people whose contract period was not for defined period had 12% lower odds of poor SRH (OR=0.88).”  

We agree that this association is not statistically significant, as the 95% CI is 0.50-1.57 (shown in table 3), therefore, changes have been made accordingly.

However, as described in lines 221-222, “the sample size of this labour market characteristic is relatively small, therefore the result should be interpreted cautiously.” There are 284 individuals in the reference group (defined period) and 299 in the not defined period group. The cause of the small sample size for this labour market characteristic is that only respondents who reported having a written-form labour contract (n=588) were asked with contract period.

Lines 223-226: The same issue as previously mentioned regarding the lines 212-220.

Changes have been made accordingly.

Table 3: In the table or under the table, it should be noted that the variable Labour contract period only the person are included who have a written labour contract.

In table 3, above the “labour contract period”, it is indicated that “respondent asked if they had a labour contract”.

Overall: One of the goals of the study was to examine the role of migrant status and of region in the association between labour market characteristics and SRH. However, these associations are not reported in the results section nor in the tables. These results should be described in the results section as well as in the additional table or tables (for example in Appendix).

The relevant description/results on roles of residence type (urban, migrant, or rural) and region are described in abstract (line 21); in section 2.5 of main text (lines 139-142); in section 4.1: main findings of the study (lines 239-243); in section 4.2 of main text (lines 271-273); and in appendix table A2 (originally in table A1 and now in table A2).

Reviewer 3 Report

This is a relevant contribution focused on the labour force characteristics and health status of the elderly population. A couple of  suggestions would improve the overall chances of it being published:

Abstract

1.     Suggest rewriting the first sentence as “The prevalence of chronic conditions increased with age……”

2.     Line 12, suggests removing ‘what kind of’

3.     Line 13, suggests rewriting as ‘data were adopted from…..’

4.     It is not clear why exclude agricultural jobs.

5.     What are ‘labour market characteristics’?

6.     It is not clear regarding the study outcome in the abstract.

7.     Suggest presenting odds ratios rather than P values.

Introduction

1.     Line 28, CVDs needs a full spell.

2.     Line 28-29, I think most people won’t have the acknowledge to make self-diagnosis, please consider rephase this sentence.

3.     I don’t quite get what do you mean about ‘ambiguous health problems’.

4.     Is the SRH a well-recognised tool for scale? What exactly it measures?

5.     Suggest changing ‘worse SRH’ to ‘poor SRH’.

6.     The introduction was mainly about SRH, and little information was provide regarding the labour market characteristics.

Materials and Methods

1.     Could the author provide more information about the CHARLS study e.g. what it’s aim is, whether is it a continued cohort study or a cross-sectional study? What is the time line for this study, and what information this study collected?

2.     What do you mean ‘living in intuition’?

3.     Figure 1, the author should clarify why farm workers were excluded.

4.     It looks like there were many labour market characteristics. Could the author provide more information about how these characteristics measured? For example, whether the ‘weekly working hours’ was self-reported in numbers or choose from multiple categories.

5.     It is not very clear about the differences between characteristics for ‘working population’ or ‘employees. It said some questions were only for employees, does this means there were questions for employers? Or do you mean questions only for people who are currently working?

6.     Regarding the SRH questions, did you ask for their health status at the moment, last month or in general?

7.     Has the author considered multiple imputations for the missing values since the percentage of missing for some key variables was quite high?

Result

1.     Does table1 show the results from chi-square tests? From my understanding, when you perform Chi-square test for multiple groups together, you will get a general idea about the differences between groups. Did you perform the sub-effect test between each groups?

2.     For the marital status, how about ‘divorced’, is this categorised into unmarried group?

3.     In the table2, nearly all ‘missing’ groups are significantly associated with poor SRH, therefore, the author should consider another technical approach to deal with missing data.

4.     Suggest reporting the 95%CI along with OR values in the text.

Discussion

1.     In the discussion section, the author mainly focused on the local content, but made less comparison with other literature. How are your findings compared with studies in other developing countries? In addition, what labour force characteristics do you think are modifiable and can be improved by interventions? Most importantly, what is the policy implications of this study?

2.     How to deal with missing values should be discussed in the limitation section.

Author Response

This is a relevant contribution focused on the labour force characteristics and health status of the elderly population. A couple of suggestions would improve the overall chances of it being published:

Abstract

Suggest rewriting the first sentence as “The prevalence of chronic conditions increased with age……”

Thank you, we have made changes as suggested.

Line 12, suggests removing ‘what kind of’

Done.

Line 13, suggests rewriting as ‘data were adopted from…..’

Done.

It is not clear why exclude agricultural jobs.

The current study focused on characteristics of Chinese non-agricultural workers. Currently there are not many labour market characteristics that we can define for farm workers. For example, working in public or private sectors, employer-provided insurances, supervisory position, labour contract, labour contract period, paid vacation or paid sick leave, and fringe benefits are not relevant for Chinese farmers. It is planned that we will look into this specific group of workers in our future research.

What are ‘labour market characteristics’?

Labour market characteristics include but are not limited to the fourteen characteristics we described in this study. It refers to a wide range of work-related conditions. We believe it is now expressed more clearly in the text.

It is not clear regarding the study outcome in the abstract.

The study outcome, as indicated in the title and abstract, is self-rated health (SRH).

Suggest presenting odds ratios rather than P values.

We agree, and the changes have been made

Introduction

Line 28, CVDs needs a full spell.

Done

Line 28-29, I think most people won’t have the acknowledge to make self-diagnosis, please consider rephase this sentence.

Done

I don’t quite get what do you mean about ‘ambiguous health problems’.

Description has been changed to make it more clear.

Is the SRH a well-recognised tool for scale?

What exactly it measures?

SRH is a well-recognised tool for measuring health status, as it was used in many longitudinal studies around the world (e.g. CHARLS, HRS, ELSA) for many years. As we originally described in lines 35-38, it “can be used to measure the general health conditions of individuals and detect some of the underlying health problems in an effective way [4]. Worse SRH is associated with higher risk of all-cause mortality in prospective studies [4-7].”

References [4-7]:

[4] DeSalvo, K. B.; Bloser, N.; Reynolds, K.; He, J.; Muntner, P. Mortality Prediction with a Single General Self-Rated Health Question A Meta-Analysis. J. Gen. Intern. Med. 2006, 20, 267-275. https://doi.org/doi.10.1111/j.1525-1497.2005.0291.x

[5] Pan, Y.; Pikhartova, J.; Bobak, M.; & Pikhart, H. Reliability and predictive validity of two scales of self-rated health in China: results from China Health and Retirement Longitudinal Study (CHARLS). BMC Public Health 2022, 22(1), 1863. https://doi.org/10.1186/s12889-022-14218-1

[6] Mossey, J. M.; Shapiro, E. Self-Rated Health: A Predictor of Mortality Among the Elderly. Am. J. Public Health 1982, 72(8), 800-808. https://doi.org/doi.10.2105/ajph.72.8.800

[7] Yu, E. S. H.; Kean, Y. M.; Slymen, D. J.; Liu, W. T.; Zhang, M.; Katzman, R. Self-perceived Health and 5-Year Mortality Risks among the Elderly in Shanghai, China. Am. J. Epidemiol. 1998, 147(9), 880-890. https://doi.org/doi.10.1093/oxfordjournals.aje.a009542

Suggest changing ‘worse SRH’ to ‘poor SRH’.

We decided not to make this change. In line with references [4-7], the “worse” here refers to less than “good”/“very good” categories, it could be “fair”, “poor”, or “very poor”, rather than the single option “poor”. Alternatively, could use the term “worse than good SRH” but find such wording rather cumbersome.

References [4-7] are given in the response to previous comment.

The introduction was mainly about SRH, and little information was provide regarding the labour market characteristics.

We agree this aspect of introduction was rather brief. More description on associations between labour market characteristics and SRH have been now added in the introduction.

Materials and Methods

Could the author provide more information about the CHARLS study e.g. what it’s aim is, whether is it a continued cohort study or a cross-sectional study? What is the time line for this study, and what information this study collected?

More description on CHARLS has been added. As we originally described in line 51, the China Health and Retirement Longitudinal Study (CHARLS) is a longitudinal study. Relevant information has been added. As we originally described in section 2 (lines 52-56): it “is a nationally representative study conducted among residents aged 45 years or older living in 28 provinces in China”, and “Information on family, health status, healthcare, work circumstances and economics status were collected. The first national survey (2011-2012) was used in the present study. Further information about CHARLS can be found elsewhere [14]”.

Reference [14] is the CHARLS cohort profile:

[14] Zhao, Y.; Hu, Y.; Smith, J. P.; Strauss, J.; Yang, G. Cohort profile: the China Health and Retirement Longitudinal Study (CHARLS). Int. J. Epidemiol. 2014, 43(1), 61-68. https://doi.org/doi.10.1093/ije/dys203.

What do you mean ‘living in intuition’?

“living in institutions” means living in nursing homes (sometimes also called retirement homes, old people’s homes). In CHARLS national baseline users’ guide, it is described as “The CHARLS sample is representative of people aged 45 and over, living in households; institutionalized mid-aged and elderly are not sampled, but Wave 1 respondents who later enter into an institution will be followed” (Section 2.2, page 15).

Reference [13] is CHARLS users’ guide:

[13] Zhao, Y.; Strauss, J.; Yang, G.; Giles, J.; Hu, P.; Hu, Y.; Lei, X.; Liu, M.; Park, A.; Smith, J. P.; Wang, Y. China Health and Retirement Longitudinal Study – 2011-2012 National Baseline Users’ Guide. National School of Development, Peking University, 2013.

Figure 1, the author should clarify why farm workers were excluded.

See our earlier response on the similar question.

It looks like there were many labour market characteristics.

Could the author provide more information about how these characteristics measured?

For example, whether the ‘weekly working hours’ was self-reported in numbers or choose from multiple categories.

14 labour market characteristics were defined and investigated in the current study, as mentioned in the abstract, methods, results, and discussion. In section 2.1 (Labour market characteristics for working population) and section 2.2 (Labour market characteristics for employees), we provided detailed information on the definition of each labour market characteristic (lines 59-106). All the data we used in the current study were collected via questionnaire in the face-to-face interviews conducted by the CHARLS team. The specific questions could be found in CHARLS national baseline questionnaire. Weekly working hours was self-reported in numbers. As we described in line 73: “Weekly working hours include hours spent on main job and additional jobs”, we calculated the total weekly working hours and categorized it.

It is not very clear about the differences between characteristics for ‘working population’ or ‘employees.

It said some questions were only for employees, does this means there were questions for employers? Or do you mean questions only for people who are currently working?

The response to this point is similar to earlier response from the other reviewer. We apologize if the distinction between two parts of the analysis was not clear enough. We have tried to clarify the distinction. There are seven labour market characteristics specific for employees and these could not be applied to the whole working population, for example, questions about labour contract, paid vacations, or fringe benefits. Relevant description has been now added. However, as we originally described in lines 130-132 “There are two groups of labour market characteristics, one group is available for the whole sample, and another group is only available for paid employees.” Therefore, “only for employees” means those characteristics are only for people who are paid employees and are not available for the whole sample. The other labour market characteristics were answered by all working individuals.

Regarding the SRH questions, did you ask for their health status at the moment, last month or in general?

SRH was asked by the question “Would you say your health is very good, good, fair, poor or very poor?. This description is included in the manuscript. There is no indicator of specific time point in the CHARLS SRH questions. Considering the interviews of CHARLS national baseline wave were face-to-face, we can hypothesize the measured health status is the general health status at the time of interview.

Has the author considered multiple imputations for the missing values since the percentage of missing for some key variables was quite high?

We agree that missingness in some variables was relatively high. We decided to keep specific category for missing to present the association in the data as they were collected.

In addition, it would be highly complex task to conduct imputations as missingness is combination of questions not being answered/answering impossible answers and questions not being asked (some questions been asked only in specific groups of subjects). When excluding those who were not asked particular questions, the missingness for not answering questions/answering with implausible answers is relatively low (between 0 -12%). It has been reported that when the proportion of missing data is under 10%, MI may only have little advantage (Barzi & Woodward, 2004). Thus in this case, MI would possibly have only little impact on findings, and we thus decided not to use MI in this manuscript.

References:

Barzi F, Woodward M. Imputations of missing values in practice: results from imputations of serum cholesterol in 28 cohort studies. Am J Epidemiol. 2004;160(1):34-45.

Result

Does table1 show the results from chi-square tests?

From my understanding, when you perform Chi-square test for multiple groups together, you get a general idea about the differences between groups. Did you perform the sub-effect test between each groups?

Indeed, chi-square tests were performed in table 1. We compared each category with the first (reference) category – this is why there are p-values in each row except those in reference categories.

For the marital status, how about ‘divorced’, is this categorised into unmarried group?

Yes, unfortunately, there was very limited group of divorced (n=50). Thus they were included in group of unmarried. Further explanations are added in the Methods.

In the table2, nearly all ‘missing’ groups are significantly associated with poor SRH, therefore, the author should consider another technical approach to deal with missing data.

This comment relates to the question on missing data and multiple imputation included above among comments on Materials and Methods; please also refer to relevant response in that section.

Suggest reporting the 95%CI along with OR values in the text.

For ORs calculated in the current study, 95% CI are reported in tables, and we thought it is not necessary to duplicate this by reporting these in the main text. We tried to refer to the appropriate results in the tables in the text. In places where we refer to ORs from other studies (references), we now report 95% Cis in most (if not all) places.

Discussion

In the discussion section, the author mainly focused on the local content, but made less comparison with other literature. How are your findings compared with studies in other developing countries?

Comparison with other studies has been added in the first paragraph of section 4.2.

In addition, what labour force characteristics do you think are modifiable and can be improved by interventions? Most importantly, what is the policy implications of this study?

More discussion on policy implications has been added in “Conclusions” section.

How to deal with missing values should be discussed in the limitation section.

Thank you. More discussion on missing data has been added in section 4.3 (Strengths and limitations).

Round 2

Reviewer 3 Report

The quality of the manuscript has been improved significantly by addressing all comments from reviewers. 

Some minor further suggestions: 

I. In the introduction, if the author could provide some statistics regarding the middle-aged labour force ( e.g. the proportion of middle-aged workers of the whole labour force, what are the leading causes of DALYs among middle age population in China), that will strengthen the rationale of the study. 

2. Table 2, Earned income, suggests to change the 'richest' to 'highest', 'poorest' to 'lowest'. In addition, was the quantile of the income estimated only apply to the sample (not considered to the general population)? 

3. I think this study captures a few labour-marked characteristics, but some important factors might be missing that could be acknowledged as limitations such as support and training provided by their employers, experience with workplace violence including bully, OHS regulations. 

4. The OR values can be removed from the text in the conclusion, but described as increased/decreased risk by xx%. 

Author Response

Dear reviewer

Thank you for this further review. We made changes to the text as marked as track changes. Here is the summary of our responses:

I. In the introduction, if the author could provide some statistics regarding the middle-aged labour force ( e.g. the proportion of middle-aged workers of the whole labour force, what are the leading causes of DALYs among middle age population in China), that will strengthen the rationale of the study. 

We added two sentences to the introductory section, one related to proportion of middle aged labour force in China, and another one related to leading causes of DALYs in Chinese population.

2. Table 2, Earned income, suggests to change the 'richest' to 'highest', 'poorest' to 'lowest'. In addition, was the quantile of the income estimated only apply to the sample (not considered to the general population)? 

We made changes in Table 2 as suggested and added explanation to the text related to quantiles being estimated from the sample.

3. I think this study captures a few labour-marked characteristics, but some important factors might be missing that could be acknowledged as limitations such as support and training provided by their employers, experience with workplace violence including bully, OHS regulations. 

We added another limitation to the discussion related to further possible labour-related characteristics unavailable in the data.

4. The OR values can be removed from the text in the conclusion, but described as increased/decreased risk by xx%. 

We decided to keep ORs in the conclusions as we found expression related to %changes in changed odds little bit confusing (some being more than 100% change) for non-statistical readers.